# Peptide Mediated Adhesion to Beta-Lactam Ring of Equine Mesenchymal Stem Cells: A Pilot Study

**DOI:** 10.3390/ani12060734

**Published:** 2022-03-15

**Authors:** Barbara Merlo, Vito Antonio Baldassarro, Alessandra Flagelli, Romina Marcoccia, Valentina Giraldi, Maria Letizia Focarete, Daria Giacomini, Eleonora Iacono

**Affiliations:** 1Department of Veterinary Medical Sciences, University of Bologna, Via Tolara di Sopra, 50, 40064 Ozzano Emilia, BO, Italy; vito.baldassarro2@unibo.it (V.A.B.); eleonora.iacono2@unibo.it (E.I.); 2Interdepartmental Center for Industrial Research in Health Sciences and Technologies, University of Bologna, Via Tolara di Sopra, 41/E, 40064 Ozzano Emilia, BO, Italy; alessandra.flagelli2@unibo.it (A.F.); romina.marcoccia.rm@gmail.com (R.M.); valentina.giraldi2@unibo.it (V.G.); marialetizia.focarete@unibo.it (M.L.F.); daria.giacomini@unibo.it (D.G.); 3IRET Foundation, Via Tolara di Sopra, 41/E, 40064 Ozzano Emilia, BO, Italy; 4Department of Chemistry “Giacomo Ciamician” and INSTM UdR of Bologna, University of Bologna, Via Selmi 2, 40126 Bologna, BO, Italy

**Keywords:** equine, mesenchymal stem cell, adhesion, α4β1 integrin, β-lactam agonist, poly L-lactic acid (PLLA) scaffold

## Abstract

**Simple Summary:**

In recent years, stem cell therapy has emerged as a promising potential treatment for chronic wounds in both human and veterinary medicine. Particularly, mesenchymal stem cells (MSCs) may be an attractive therapeutic tool for regenerative medicine and tissue engineering because these cells play a critical role in wound repair and tissue regeneration due to their immunosuppressive properties and multipotency. The use of biomaterials with integrin agonists could promote cell adhesion increasing tissue repair processes. This pilot study focuses on the adhesion ability of equine adult (adipose tissue) and fetal adnexa (Wharton’s jelly) derived MSCs mediated by GM18, an α4β1 integrin agonist, alone and combined with a biodegradable polymeric scaffold. Results show that a 24 h exposition to soluble GM18 affects equine MSCs adhesion ability with a donor-related variability and might suggest that WJ-MSCs more easily adhere to poly L-lactic acid (PLLA) nanofibers combined with GM18. These preliminary results need to be confirmed by further studies on the interactions between the different types of equine MSCs and GM18 incorporated PLLA scaffolds before drawing definitive conclusions on which cells and scaffolds could be successfully used for the treatment of decubitus ulcers.

**Abstract:**

Regenerative medicine applied to skin lesions is a field in constant improvement. The use of biomaterials with integrin agonists could promote cell adhesion increasing tissue repair processes. The aim of this pilot study was to analyze the effect of an α4β1 integrin agonist on cell adhesion of equine adipose tissue (AT) and Wharton’s jelly (WJ) derived MSCs and to investigate their adhesion ability to GM18 incorporated poly L-lactic acid (PLLA) scaffolds. Adhesion assays were performed after culturing AT- and WJ-MSCs with GM18 coating or soluble GM18. Cell adhesion on GM18 containing PLLA scaffolds after 20 min co-incubation was assessed by HCS. Soluble GM18 affects the adhesion of equine AT- and WJ-MSCs, even if its effect is variable between donors. Adhesion to PLLA scaffolds containing GM18 is not significantly influenced by GM18 for AT-MSCs after 20 min or 24 h of culture and for WJ-MSCs after 20 min, but increased cell adhesion by 15% GM18 after 24 h. In conclusion, the α4β1 integrin agonist GM18 affects equine AT- and WJ-MSCs adhesion ability with a donor-related variability. These preliminary results represent a first step in the study of equine MSCs adhesion to PLLA scaffolds containing GM18, suggesting that WJ-MSCs might be more suitable than AT-MSCs. However, the results need to be confirmed by increasing the number of samples before drawing definite conclusions.

## 1. Introduction

Wound healing is a very complex physiological response to the disruption in the normal architecture of the skin and is influenced by many factors [1]. This is a stepwise process that includes a proliferative phase, in which damaged tissue is removed and granulation tissue forms in the wound, and a remodeling phase, with the formation of scar tissue indicating the completion of the wound healing process [2]. The conventional approaches applied for the instant healing of skin wounds include the use of different drugs and natural products with anti-inflammatory, anti-microbial, and antioxidant properties [3].

In recent years, wound healing has become a logical target for innovative therapies, such as regenerative medicine strategies, which have the potential to restore tissue, perhaps equaling or exceeding pre-damage levels, resulting in improved outcomes and quality of life [4]. Stem cell therapy has emerged as a promising potential treatment for chronic wounds [5]. Particularly, mesenchymal stem cells (MSCs) may be an attractive therapeutic tool for regenerative medicine and tissue engineering because they play a critical role in wound repair and tissue regeneration due to their immunosuppressive properties and multipotency [6]. 

Regenerative medicine applied to skin lesions has been a field of constant improvement for both human and veterinary medicine [7,8,9,10]. The objective of regenerative medicine is to stimulate the self-repair of tissues and organs with stem cells alone or in conjunction with biomaterials [11]. Biomaterials have a very important role in tissue engineering but, unlike natural polymers, synthetic polymeric biomaterials used in tissue engineering applications lack biological activity and typically do not promote excellent cell adhesion and growth [12,13]. Cell–cell and cell–extracellular matrix (ECM) interactions in the niche are mediated by different cell adhesion molecules, and integrins are one of the main players [14].

Integrins are transmembrane receptors comprised of two subunits, alpha (α) and beta (β). The molecular family includes 18 α and 8 β subunits, leading to the formation of 24 different heterodimeric transmembrane receptors [15,16]. They mediate cellular interactions with the ECM and surrounding cells by binding specific ligands [14,16] regulating crucial aspects of cellular functions, including adhesion, differentiation, growth, gene expression, and survival [17]. The ability of integrins to bind and associate with various soluble ligands largely depends on the structural conformations of the α and β subunits.

To modulate integrins’ action, a novel series of β-lactam targeting RGD (arginine–glycine–aspartic acid) fragment and leukocyte integrins were designed [18]. β-lactams have two specific structural features that are of interest with regard to biological activity: a constrained four-membered cyclic amide, which could easily undergo ring-opening reactions by nucleophilic residues in the active sites of enzymes, and a rigid core structure that, by reducing the conformational degrees of freedom, could favor and actually enhance directional noncovalent bonding for ligand–receptor recognition [19]. On their feature, Baiula et al. [18] obtained selective and potent agonists that could induce cell adhesion and promote cell signaling mediated by αvβ3, αvβ5, α5β1, or α4β1 integrin, and antagonists for the integrins αvβ3 and α5β1, as well as α4β1 and αLβ2, preventing the effects elicited by the respective endogenous agonists [18].

Electrospinning is a highly versatile method to process polymer solutions or melts into continuous fibers in the form of non-woven porous mats that find application in the biomedical field where they are used as tissue engineering scaffolds [20,21]. The use of biomaterials, such as electrospun scaffolds, conjugated with integrin agonists could promote cell adhesion by influencing tissue repair processes and therapeutic progress. The modulation of the activity of integrins can represent an excellent strategy applicable to cell therapies that aim to stimulate tissue repair using MSCs.

New scaffolds based on electrospun poly L-lactic acid (PLLA) and agonist ligands of monocyclic β-lactam compounds of specific integrins have recently been tested [22]. Incorporation of GM18 β-lactam into PLLA scaffolds has been shown to support increased cell proliferation of human MSCs from bone marrow [22]. Studies on the use of scaffolds with integrin agonists based on β-lactams in equines have never been reported. Therefore, the aim of this pilot study was to analyze the effect of an α4β1 integrin agonist on cell adhesion of equine adult (adipose tissue; AT) and fetal membrane (Wharton’s jelly; WJ) derived MSCs and to preliminarily investigate the adhesion of these cells to GM18-incorporated PLLA scaffolds, to evaluate their potential use for the treatment of decubitus ulcers.

## 2. Materials and Methods

### 2.1. Materials

Poly L-Lactic Acid (PLLA) (Lacea H.100-E Mw 8.4 × 104 g/mol) was purchased from Mitsui Fine Chemicals (Duesseldorf, Germany). Dichloromethane (DCM) and dimethylformamide (DMF) were purchased from Sigma–Aldrich (Milan, Italy) and used without further purification. β-Lactam GM18 was prepared accordingly to a previously reported multistep synthesis [18]. Other chemicals were purchased from Sigma–Aldrich, culture media from Life Technologies (Monza, Italy), and laboratory plastic was from Sarstedt Inc. (Verona, Italy) unless otherwise stated. 

### 2.2. Fabrication of GM18-Incorporated PLLA Scaffolds

Scaffolds were fabricated using a homemade electrospinning apparatus, consisting of a high-voltage power supply (Spellman SL 50 P 10/CE/230), a syringe pump (KD Scientific 200 series), a glass syringe containing the polymer solution and connected to a stainless steel blunt-ended needle (inner diameter = 0.51 mm) through a PTFE tube. Electrospinning was performed at room temperature (RT) and with a relative humidity of 50−60%. Blends of the β-lactam compound and the polymer were prepared by dissolving the two components in a mixed solvent of DCM:DMF = 65:35 *v/v* at a polymer concentration of 13% *w/v* and a concentration of β-lactam of 0, 5, 10, and 15 wt% with respect to the polymer. The polymeric solutions were electrospun by applying the following processing conditions: applied voltage = 22 kV, feed rate = 1 mL/h, needle-to-collector distance = 15 cm. The scaffolds were dried on P_2_O_5_ under vacuum for three days to remove any solvent residue, sterilized by irradiation with γ rays, and finally stored at 4 °C.

### 2.3. Animals

Intra-abdominal AT was collected from horses during colic surgery (*n* = 3) and umbilical cord (UC) samples (*n* = 3) were collected after physiological birth. All owners spontaneously referred the animals to the Department of Veterinary Medical Sciences (University of Bologna) and gave written consent to allow for the use of removed tissue for research purposes. Experimental procedures were approved by the Ethics Committee on animal use of the University of Bologna (Prot. 55948-X/10). 

### 2.4. Sample Collection and Cell Isolation 

AT and UC samples were stored in DPBS (Dulbecco’s Phosphate Buffered Saline) supplemented with antibiotics (100 IU/mL penicillin, 100 μg/mL streptomycin) immediately after removal, and kept at 4 °C until the transfer to the lab. Under a laminar flow hood, the richest portion of WJ was immediately isolated from the cord tissue. For both tissues, MSCs were isolated as previously described [23]. Briefly, samples were washed by repeated immersion in DPBS, weighed, and cut into 0.5 cm pieces using sterile scissors. Minced tissue was transferred into a 50 mL polypropylene tube and processed by enzymatic digestion using a 0.1% collagenase type I solution in DPBS (1 mL solution/1 g tissue). The suspension was vigorously mixed every 10 min while kept in a 37 °C water bath for an overall 30 min period. Then, collagenase was inactivated by diluting the suspension 1:1 with DPBS supplemented with 10% FBS. The resulting solution was filtered through a stainless-steel mesh to discard the undigested tissue and centrifuged at 470× *g* for 10 min at 25 °C. The cell pellet was suspended in a culture medium consisting of DMEM/MEM 1:1, plus 10% FBS and antibiotics (100 U/mL penicillin, 100 μg/mL streptomycin). Cells were plated as “Passage 0” (P0) in a 25 cm^2^ flask containing 5 mL of culture medium and cultured in humidified air with 5% CO_2_ at 38.5 °C. After 48 h of in vitro culture, non-adherent cells were removed by completely re-placing the culture medium. Then cell culture medium was changed twice a week until cell growth reached 80 to 90% confluence. 

### 2.5. Cell Freezing and Thawing 

In order to perform all tests at the same time for all samples, reducing experimental/technical effects, P0 cells were cryopreserved and stored in liquid nitrogen. AT- and WJ-MSCs were deep-frozen as previously described [24]. Briefly, when cells reached 80 to 90% confluence, they were trypsinized (0.25% trypsin) for 10 min. Then DPBS plus 10% FBS was added 2:1 to inactivate trypsin and the cell suspension was centrifuged at 470× *g* for 10 min at 25 °C. The pellet was suspended in 0.5 mL of FBS and transferred into a 1.5 mL cryogenic tube kept at 5 to 8 °C for 10 min. Refrigerated cells were diluted 1:1 with FBS plus 16% DMSO (dimethyl sulfoxide) reaching a final concentration of 8% DMSO. The suspension was further kept at 5 to 8 °C for 10 min, then the cryogenic tube was put in Mr Frosty (Nalgene) at −80 °C for 24 h and finally stored in liquid nitrogen. For thawing, AT- and WJ-MSCs vials were immersed in a water bath at 37 °C, the suspension was transferred into a 50 mL tube and dropwise diluted with 20 mL of culture medium, then centrifuged at 470× *g* for 10 min at 25 °C. The pellet was suspended in 1 mL of culture medium and cell concentration was evaluated by using a hemocytometer. Cells were plated in a 25 cm^2^ flask (5000 cells/cm^2^) as “Passage 1” (P1). Cells were allowed to proliferate until 80 to 90% confluence before trypsinization and successive passage. 

### 2.6. Growth Curve

At Passage 3 (P3) of in vitro culture, the effect of GM18 on the growth capacity of AT- and WJ-MSCs was evaluated with a growth curve. After trypsinization, nucleated cells were centrifuged at 470× *g* for 10 min at 25 °C and the pellet was suspended in 1 mL of culture medium. Cell concentration was evaluated by using a hemocytometer and 5000 cells/cm^2^ were plated in 35 mm Petri dishes and cultured for 6 days in presence of different concentrations (0 µg/mL, 5 µg/mL, 10 µg/mL, 20 µg/mL) of GM18 solubilized with DMSO. Cells were trypsinized every 24 h and the concentration was evaluated by using a hemocytometer. The same procedure was repeated up to 120 h. Cell doubling number (CD) was calculated according to the following formula:CD = ln (Nf/Ni)/ln (2)
where Nf and Ni are the final and the initial number of cells, respectively.

The experiment was done for all donors using three replicate wells for each treatment at each time point.

### 2.7. Adhesion Assay with GM18 Coating

GM18 was solubilized with DMSO and then different concentrations (0 µg/mL, 5 µg/mL, 10 µg/mL, 20 µg/mL) of GM18 were prepared with PBS. GM18 dilutions were placed into a 24-multiwell plate and incubated under a laminar flow hood to form the coating. After removing the supernatant, AT- and WJ-MSCs were added to the wells (10,000 cells/well) and incubated at 37 °C for 20 min, 2 h, 4 h, and 6 h. At the end of this time, the AT- and WJ-MSCs were fixed with 4% paraformaldehyde (PAF 4%) and allowed to incubate for 30 min at RT. MSCs were then stained with a nuclear dye (Hoechst 33,258) and the 24-multiwell plates were analyzed using a cell-based high-content screening (HCS). The experiment was done for all donors using three replicate wells for each treatment at each time point.

### 2.8. Adhesion Assay with Soluble GM18 in Pre-Treatment

GM18 was solubilized and diluted as described above. In order to investigate the influence of soluble GM18 on the subsequent cell adhesion ability, a 24 h pre-treatment was considered suitable to stimulate a cell response. AT and WJ-MSCs were initially seeded in 35 mm Petri dishes (5000 cells/cm^2^) and cultured for 24 h to allow for standard cell adhesion on plastic. Then, different concentrations of GM18 (0 µg/mL, 5 µg/mL, 10 µg/mL, and 20 µg/mL) were added to the culture medium. After 24 h of in vitro culture in presence of soluble GM18, MSCs were detached and passed into a 24-multiwell plate (10,000 cells/well) and incubated at 38.5 °C for 20 min, 2 h, 4 h, and 6 h. At the end of the incubation period, the AT- and WJ-MSCs were fixed with PAF 4% at RT for 30 min. MSCs were stained with a nuclear dye (Hoechst 33,258) and the multiwell plates were analyzed using a HCS. The experiment was done for all donors using three replicate wells for each treatment at each time point.

### 2.9. Adhesion Assay on Scaffolds

Four different scaffolds were used for this assay: PLLA without GM18; PLLA + GM18 5%; PLLA + GM18 10%; PLLA + GM18 15%. The sterile scaffolds were cut and included inside rings that allowed them to be locked inside a 24-multiwell plate. Circular cover glasses were used as control supports for cell detection analysis. Scaffolds were pre-wetted by adding 0.5 mL of 30% ethanol in DPBS for 2 s, then washed twice with 0.5 mL of DPBS for 5 min. AT- and WJ-MSCs were plated on the mounted scaffolds and incubated in humidified air with 5% CO_2_ at 38.5 °C for 20 min and 24 h. At the end of the incubation, the MSCs were fixed with PAF 4% for 30 min at RT and stained with a nuclear dye (Hoechst 33,258) and the 24-multiwell plates were analyzed using HCS. The experiment was done using three replicate wells for each treatment at each time point. Only samples that showed downregulation of integrin genes after 24 h exposure to GM18 (CV 6-20 and CV 6-15) were chosen for these experiments.

### 2.10. Cell-Based High Content Screening Analysis

For all the adhesion assays, the HCS technology (Cell Insight NXT, Thermo Scientific, Waltham, MA, USA) was used to detect and count all the cells present in each analyzed well. Analysis was performed by selecting the general intensity measurement assay (Compartmental analysis) from the software algorithm list (HCS Studio v. 6.6.0, Thermo Scientific). Using nuclear staining, cells were detected and counted in each well of the whole cultures.

### 2.11. Cell Viability Assay

A viability test was performed to test the effect of DMSO, used to solubilize GM18, on equine MSCs. Cells were cultured for 24 h in standard medium, then DMSO was added at the 3 concentrations used for GM18 dilutions. After 24 h of culture, cell viability was evaluated by staining cells with 0.05% Eosin. 

### 2.12. Molecular Characterization 

To analyze the effects of GM18 on gene expression, MSCs were cultured in vitro for 24 h with a culture medium supplemented with GM18 (0 μg/mL, 5 μg/mL, 10 μg/mL, 20 μg/mL) and then lysed.

#### RNA Extraction, Reverse Transcription, and qPCR

The RNeasy Micro Kit (QIAGEN, Milan, Italy) was used for the total RNA extraction, then quantified with Nanodrop 2000 spectrophotometer (Thermo Scientific). First-strand cDNA was produced using the iScript gDNA Clear cDNA Synthesis Kit (BioRad, Hercules, CA, USA). An RNA sample with no reverse transcriptase enzymes in the reaction mix was processed as a no-reverse transcription control sample. Semi-quantitative real-time PCR reactions were performed in a final volume of 20 μL (1× SYBR Green qPCR master mix—BioRad—and 0.5 μM forward and reverse primers), using the CFX96 real-time PCR system (BioRad). To test possible genomic DNA contaminations, the no-reverse transcriptase sample (NoRT) was processed in parallel with the others.

Details of the primer sequences are included in Table 1.

PCR reactions were performed using the following steps and thermal profile: denaturation step (98 °C for 3 min) and amplification (95 °C for 10 s and 60 °C for 60 s; 40 cycles), followed by the melting curve of the amplified products (55 °C to 95 °C, ΔT = 0.5 °C/s).

Primer efficiency values for all primers were 95–102%; therefore the 2 ^(^^−ΔΔCt)^ method was used to perform the analysis.

### 2.13. Statistical Analysis

Data are reported as mean ± SEM. Prism software (v.9; GraphPad Software, San Diego, CA, USA) was used for statistical analyses and graph generation. Data were collected from at least three independent experiments. Two-Way ANOVA was used for the population doubling, while One-Way ANOVA was used for all the other analyses. The results were considered significant when the probability of their occurrence as a result of chance alone was <5% (*p* < 0.05).

## 3. Results

### 3.1. GM18 Treatment Affects the Adhesion of Adipose Tissue-Derived Mesenchymal Stem Cells

We analyzed the effect of the integrin α4β1 ligand GM18 on cells derived by adult equine AT, isolated from three different animals (CV6-20, CV4-20, and CV24-19). We used the adhesion assay, counting by HCS the cells present in three replicate wells at specific times after the cell seeding, from two different conditions: using GM18 as a coating (Figure 1A,C,E), or pre-treating the cells for 24 h with GM18, and after detaching and seeding back on 24 multiwell plates (Figure 1B,D,F).

The GM18 coating only showed a significant increase in adhesion in CV6-20 derived cells cultures after 2 h at all the analyzed doses (One Way ANOVA, F(3,8) = 23.34; *p* = 0.0003; followed by Dunnett’s post-test; 5 µg/mL, *p* = 0.0018; 10 µg/mL, *p* = 0.0060; 20 µg/mL, *p* = 0.0001) (Figure 1A).

An increase in adhesion for all the three cultures preparation at different time points was detected after GM18 was in solution 24 h and cells were detached and analyzed for their adhesion capacity.

In particular, cells derived from CV6-20 showed an increased adhesion at 2 h (One Way ANOVA, F(3,8) = 47; *p* < 0.0001), 4 h (One Way ANOVA, F(3,8) = 21.40; *p* = 0.0004), and 6 h (One Way ANOVA, F(3,8) = 9.245; *p* < 0.0056) from seeding, when treated with 10 µg/mL (Dunnett’s post-test; 2 h, *p* < 0.0001; 4 h, *p* = 0.0005; 6 h, *p* = 0.0046). At 6 h, a 20 µg/mL dose was also effective in increasing the adhesion (*p* = 0.0329) (Figure 1B).

The cultures isolated from CV4-20 respond at the 24 h treatment with an increased adhesion in a short time (20 min; One-Way ANOVA, F(3,8) = 15.18; *p* = 0.0012; Dunnett’s post-test, 5 µg/mL, *p* = 0.0004; 20 µg/mL, *p* = 0.0137) (Figure 1D).

In addition, the cultures produced from CV24-19 showed an increase in adhesion at 20 min (One-Way ANOVA, F(3,8) = 7.934, *p* = 0.0088, 10 µg/mL *p* = 0.0250) but also at the longer time points, at 2 h (One-Way ANOVA, F(3,8) = 10.14, *p* = 0.0042) and 4 h (One-Way ANOVA, F(3,8) = 5.704, *p* = 0.0219), at the highest dose (Dunnett’s post-test; 20 µg/mL; 2 h, *p* = 0.0140; 4 h, *p* = 0.0272) (Figure 1F).

All data were corrected by the effect of the vehicle analyzed by the viability test (data not shown). 

Representative panels of representative images from statistically significant results are included in Appendix A.

### 3.2. GM18 Treatment Impacts on the Adhesion of Wharton’s Jelly Derived Mesenchymal Stem Cells

The same protocol was used for the analysis of the GM18 effect on cells isolated from WJ (Figure 2). Further, in this case, the coating was not effective in influencing the cell adhesion, even if a single significant increase was described for cells isolated from CV6-15, at 20 min (One-Way ANOVA, F(3,8) = 5.461, *p* = 0.0245) and the lowest dose (Dunnett’s post-test, 5 µg/mL, *p* = 0.0104) (Figure 2C).

For these cells, the effect of the pre-treatment seems to induce both an increase or decrease in adhesion, depending on the origin of cells and the analyzed time.

In fact, for CV12-15 (Figure 2B), with at early time point (20 min, One-Way ANOVA, F(3,8) = 0.0185, *p* = 6.075) at highest dose (20 µg/mL, Dunnett’s post-test, *p* = 0.0074), the GM18 exposure generated an increase in cell adhesion, while it was decreased at all the other analyzed times (One-Way ANOVA, 2 h, F(3,8) = 0.0001; *p* = 29.09; 4 h, F(3,8) = 7.939, *p* = 0.0088; 6 h, F(3,8) = 12.95, *p* = 0.0003) at 5 µg/mL (Dunnett’s post-test, 2 h, *p* = 0.0003; 4 h, *p* = 0.0205; 6 h, *p* = 0.0008) and 10 µg/mL (2 h, *p* = 0.0005; 4 h, *p* = 0.0381; 6 h, *p* = 0.0046).

CV6-15 cells, instead, showed only an increase in adhesion at 2 and 4 h (One-Way ANOVA, 2 h, F(3,8) = 6.279, *p* = 0.0169; 4 h, F(3,8) = 4.745, *p* = 0.0348) at the highest concentration (Dunnett’s post-test, 20 µg/mL, 2 h, *p* = 0.0073; 4 h, *p* = 0.0237) (Figure 2D). For the CV3-15 samples (Figure 2F), the effect favored the adhesion at 2 h with the highest dose (One-Way ANOVA, F(3,8) = 209.5, *p* < 0.0001; Dunnett’s post-test, 20 µg/mL *p* < 0.0001), and at 4 h (One-Way ANOVA, F(3,8) = 29.46, *p* = 0.0001) and 6 h (F(3,8) = 23.22, *p* = 0.0003) at 10 µg/mL (Dunnett’s post-test, 4 h, *p* = 0.0085; 6 h, *p* = 0.0348). However, the use of the highest concentration in these cells reduced the adhesion (20 µg/mL, 4 h, *p* = 0.0028; 6 h, *p* = 0.0025).

Representative panels of representative images from statistically significant results are included in Appendix A.

### 3.3. GM18 Effect Is Highly Variable between Cell Isolation from Different Donors

Following the analysis of the individual data produced by treatment of cells isolated from different animals, we pooled the data from different isolations. Both for AT (Figure 3A,B) and WT (Figure 3C,D) derived cell, and for both protocols (coating or soluble treatment), the cells exposed to the molecule resulted in a highly variable response.

Due to this high variability, it is not possible to describe any statistically significant effect of the GM18 exposure. However, the results were significant (One-Way ANOVA, F(3,32) = 7.721, *p* = 0.0005) at 20 µg/mL (Dunnett’s post-test, *p* = 0.0005) only for soluble treatment in WJ derived cells, analyzed after 2 h from the seeding.

### 3.4. GM18 Exposure Deregulates the Gene Expression of the Target Integrins

Of the three samples analyzed for both AT and WJ, for two of them, exposure to GM18 for 24 h did not result in gene expression regulation of the target integrins (Appendix A). However, for cells isolated from CV6-20 (AT) and CV6-15 (WJ), the exposure to the molecule resulted in the deregulation of the target genes. For these two samples, all the concentrations were tested (Figure 4).

MSCs of the CV6-20 showed a reduction in the expression of *ITGB1* (One-Way ANOVA, F(3,8) = 39.24, *p* < 0.0001) and *ITGA4* (One-Way ANOVA, F(3,8) = 206.7, *p* < 0.0001) genes at all the concentrations tested (Dunnett’s post-test, Itgb1, 5 µg/mL, *p* = 0.0006; 10 µg/mL, *p* = 0.0002; 20 µg/mL, *p* < 0.0001; Itga4, 5 µg/mL, *p* < 0.0001; 10 µg/mL, *p* < 0.0001; 20 µg/mL, *p* < 0.0001).

For the cells isolated from WJ of CV6-15, the treatment generated a reduction in the expression of both integrin subunits (One-Way ANOVA, *ITGB1*, F(3,8) = 6.017, *p* = 0.0237; *ITGA4*, F(3,8) = 12.11, *p* = 0.0037). However not all doses were effective. For *ITGB1* expression, the reduction was caused by the treatment with a concentration of 10 µg/mL (Dunnett’s post-test, *p* = 0.0155) and 20 µg/mL (*p* = 0.0202), while for *ITGA4*, only the highest dose was effective (*p* = 0.0106).

### 3.5. GM18 Exposure Affects the Population Doubling

To investigate if the effect on the cell adhesion and gene expression may affect the cell division, we analyzed the population doubling of each cell preparation (Figure 5). For the AT, cells derived from CV6-20 and CV24-19 showed an effect related to drug concentration (Two-Way ANOVA, CV6-20, F(3,8) = 5.042, *p* = 0.0299; CV24-19, F(3,8) = 5.747, *p* = 0.0214). For WJ derived cells, all samples were significantly affected by the presence of the GM18 in the culture medium (Two-Way ANOVA, CV6-15, F(3,8) = 4.068, *p* = 0.0500; CV3-15, GM18 concentration, F(3,8) = 17.53, *p* = 0.0007; CV12-15, F(3,8) = 12.42, *p* = 0.0022).

Moreover, an interaction effect between time and concentration was reported for the same cell preparations which resulted more affected by GM18: one sample of the AT (CV6-20; F(15,14) = 2.314, *p* = 0.0175) and one sample of the WJ (CV6-15; F(15,14) = 1.939, *p* = 0.0481).

The Dunnett’s post-test revealed differences at single time points for AT in the sample CV24-19 (day 4, 5 µg/mL, *p* = 0.0203; day 5, 5 µg/mL, *p* = 0.0145) and for WJ in the sample CV3-15 (day 2, 5 µg/mL, *p* = 0.0086; 10 µg/mL, *p* = 0.0298).

Analyzing data from AT and WJ pooled samples (Figure 6), the statistical analysis revealed a strong effect of time, GM18 concentration, and the interaction between the two variables in AT-MSCs (Two-way ANOVA, time, F(2.480,79.36) = 182.3, *p* < 0.0001; GM18 concentration, F(3,32) = 6.523, *p* = 0.0014; interaction, F(15,160) = 5.278, *p* < 0.0001), while only the effect of time and concentration for WJ-MSCs (Two-way ANOVA, time, F(2.884,92.29) = 346.2, *p* < 0.0001; GM18 concentration, F(3,32) = 3.418, *p* = 0.0289). For AT derived MSCs there are time point differences at day 1 (Dunnett’s post-test, 10 µg/mL, *p* = 0.0082) and day 5 (5 µg/mL, *p* = 0.0112), while for WJ derived MSCs, there were differences only at longer time points, namely at day 5 (20 µg/mL, *p* = 0.0158) and day 6 (20 µg/mL, *p* = 0.0043).

### 3.6. PLLA Scaffolds Containing GM18

The two types of cells, AT cells and WJ derived cells, which were more sensitive to GM18 treatments, both as adhesion assay and gene expression regulation (CV6-20 for AT and CV6-15 for WJ), were used to investigate the effect of a PLLA scaffold containing different concentrations of GM18 on the cell adhesion (Figure 7). Cultures seeded on scaffolds containing GM18 were compared to cells cultured on PLLA control scaffolds. The method was also tested on standard cultures seeded on glass (Appendix A).

For cells derived from AT (CV6-20), the presence of GM18 did not affect the adhesion capacity, mainly due to a high variability detected at a concentration of 10% (Figure 7A). Moreover, analyzing the differences in the cell number between the two analyzed time points, the cultures seeded on the PLLA control scaffold were not able to increase the cell number after 24 h, reflecting an impairment in the early proliferation (Student’s *t*-test = 0.0641). However, the presence of GM18 in all the analyzed concentrations was able to induce an increase in cell number after one DIV (Student’s *t*-test, 5%, *p* = 0.0170; 10%, *p* = 0.0144; 15%, *p* = 0.0395) (Figure 7A).

For the cultures obtained from WJ (CV6-15), the cell adhesion was affected after 24 h (One-way ANOVA, F(3,8) = 4.671; *p* = 0.0427), with a drastic increase due to the highest concentration of GM18 (Dunnett’s posttest, *p* = 0.0368). The same concentration was also the only condition producing an increase in cell number after one DIV (Student’s *t*-test, *p* = 0.0021) (Figure 7B).

Representative images of the cultures are included in Figure 7C,D, showing the morphology of the cultures with different magnifications (10× or 4×). 

To validate the analysis, control cells cultured on glass coverslips at the same time points were included (Appendix A), also proving the capacity of the HCS software to visualize and detect the nuclei, at low magnification objective, with a correct segmentation algorithm identifying the single nuclei (Appendix A). 

## 4. Discussion

The number of investigations into the interaction of MSCs with integrin α4β1 ligands is still limited. The agonist used in this pilot study, GM18, was designed with an amine, a carboxylate side chain, and the β-lactam ring as a site of conformational restriction, to assure the necessary integrin affinity and selectivity by improving the alignment on the receptor [18]. Among different β-lactams, GM18 was the best at enhancing the adhesion of human bone marrow-derived MSCs to PLLA electrospun scaffolds [22]. In another research, a high-affinity integrin α4β1 ligand was successfully tested on human chorionic villus-derived MSCs [25]. The adherence of cells to polymeric electrospun scaffolds increased when the surface was treated with the ligand [25]. 

Here we present the first preliminary results about the effects of an integrin α4β1 ligand on equine MSCs. GM18’s influence on the adhesion ability of both adult (AT) and fetal membrane (WJ) derived MSCs was studied when used as a coating or as a pre-treatment before standard seeding. Furthermore, the potential use of GM18 to increase cell adhesion to a PLLA scaffold was investigated from the perspective of its application for the treatment of decubitus ulcers. Significant results were observed, despite the limitations of a pilot study and the related critical issues, emerging from the donor variability in a relatively low number of samples. Nonetheless, these findings represent a fundamental phase that seems to validate the feasibility of similar studies at a larger scale.

In the horse, a few studies about α4β1 integrin are available [26,27,28,29,30,31,32], and are related to its presence on leukocytes [26,29] and endometrial epithelial cells [27,28] or equine herpesvirus 1 infection mechanism [30,31,32]. Integrins represent one of the most important families of cell adhesion receptors that mediate cell–cell and cell–ECM interactions. α4β1 integrin (also known as CD49d/CD29 or very late antigen-4, VLA-4) plays a crucial role in inflammatory diseases, cancer development, metastasis, and stem cell mobilization or retention [33]. It is involved in the regulation of hematopoietic stem cell homing and retention within the bone marrow niche [34] and its activation may be a promising strategy to improve cell retention and engraftment in stem cell-based therapies [35].

There is a wide range of MSCs, which can be isolated from different tissues. So far, donor-to-donor variations can be present even when MSCs are derived from the same tissue [36,37]. This is confirmed in the present study, where the effect of GM18 on both equine AT- and WJ-derived MSCs was highly variable in relation to the different donors. Even if only three donors for each tissue were used in this pilot study, the number was sufficient to preliminarily corroborate the hypothesis of donor variability and to highlight the critical issues of testing molecules and scaffolds with biological materials, such as MSCs. When observing the pooled data from AT-MSCs samples, it was not possible to detect a significant effect of GM18 on the cell adhesion ability. On the other hand, when considering results from each donor, it was evident that the GM18 coating increased adhesion only in one sample (CV6-20) after 2 h, while soluble GM18 was able to improve adhesion in the other samples. For WJ-MSCs, a positive effect of soluble GM18 at 20 μg/mL after 2 h was also observed in pooled data. As for AT-MSCs, the GM18 coating increased adhesion only in one WJ-MSCs sample (CV6-15), while soluble GM18 increased cell adhesion for all donors. However, differently from AT-MSCs, a decrease in cell adhesion was also observed in the other two samples at different times and GM18 concentrations. Overall, these preliminary results suggest that GM18 affects MSCs’ adhesion ability also in the horse and that a donor-based and a tissue-based response should be considered when selecting samples for cell-based therapies. 

Gene expression analysis for target integrins confirmed what was observed by the adhesion assays. In fact, only in the samples that were more influenced in vitro by the presence of GM18 was it possible to observe a reduction in the expression of both integrin subunits. While AT-MSCs were sensible to all agonist concentrations, WJ-MSCs appeared to require a higher GM18 dosage to downregulate α4β1 integrin genes. 

The population doublings of cells were also affected by the presence of GM18 over time. On the whole, a higher number of doublings was observed for AT-MSCs on day 1, with 10 μg/mL of GM18, and on day 5, with 5 μg/mL of GM18, while for WJ-MSCs, the number of doublings was higher on day 5 and 6 with the highest concentration of agonist (20 μg/mL of GM18), confirming adhesion and gene expression findings. Integrins have been identified as regulators of mitotic events [38]. It has been demonstrated that the β1 integrin is involved in the orientation of the mitotic spindle and adhesion in basal epidermal cells [39] and HeLa cells [40]. Downregulation of the β1 integrin resulted in severe misorientation of spindles [40]. In nonpolarized adherent cells, the β1 integrin-dependent mechanism for spindle orientation may ensure the attachment of both daughter cells to the substratum after cell division and prevent their mitosis-associated detachment [40]. These findings might explain the behavior of equine MSCs growth curves in presence of an integrin α4β1 ligand, especially for those cells where the β1 subunit gene expression was more deregulated.

PLLA scaffolds containing different concentrations of GM18 were seeded with the same two samples that were more sensitive to GM18 treatments and cell adhesion was evaluated. It was evident that equine AT-MSCs’ adhesion ability to PLLA scaffold was not influenced by the presence of GM18. However, the presence of GM18 increased cell adhesion after 24 h of culture. WJ-MSCs seem more suitable for PLLA scaffold adhesion, and the presence of 15% GM18 increased cell adhesion at 24 h. As the results are derived from only one suitable sample for each tissue, these preliminary results need to be confirmed before drawing definite conclusions. Research on equine MSCs and PLLA scaffolds is aimed at osteogenic differentiation. It has been observed that the osteogenic differentiation capacity of equine adipose tissue-derived MSCs on a nano-bioactive glass-coated PLLA nanofibers scaffold was higher than on the uncoated PLLA scaffold [41]. In another study using equine MSCs from adipose tissue and bone marrow, the addition of minerals to polymer scaffolds enhanced equine MSC osteogenesis over polymer alone [42]. Similarly, coating PLLA scaffolds with zinc silicate mineral nanoparticles improved in vitro osteogenic differentiation of equine adipose tissue-derived MSCs compared to uncoated PLLA scaffold [43]. The present work is the first report of equine WJ-MSCs cultivation on PLLA scaffolds. Human WJ-derived MSCs were seeded into polyglycolic acid (PGA) and PLLA scaffolds to investigate the potential ability of chondrogenesis in vitro [44]. The adhesion rate of human WJ-MSCs on PLLA scaffolds was 58 ± 6% and 75 ± 4% 3 and 6 h after seeding, respectively [44]. The same GM18 coated PLLA scaffolds used in the present study were tested with human bone marrow-derived MSCs and GM18 increased the rate of cell adhesion after 2 h of co-incubation [22]. It might be speculated that for equine AT-MSCs a longer co-incubation is needed than for WJ-MSCs in order to achieve higher adhesion rates. Nevertheless, further studies are necessary to better assess equine AT- and WJ-MSCs’ adhesion ability on PLLA scaffolds over time and to test the concurrent effect of the presence of GM18. 

## 5. Conclusions

The α4β1 integrin agonist GM18 affects equine AT- and WJ-derived MSCs’ adhesion capability. Soluble GM18 pre-treatment was more effective in enhancing cell adhesion than GM18 coating. However, there was a high variability related to different donors and it should be carefully considered when selecting samples for cell-based therapies. Samples more responsive to GM18 showed a downregulation of the target integrins’ genes and an effect on cell divisions. Preliminary results on equine MSCs adhesion to PLLA scaffolds containing GM18 might suggest that WJ-MSCs are more suitable than AT-MSCs for the development of cell-based scaffolds to be used for wound healing, but additional research is required to examine in depth the various cell characteristics after seeding and prolonged culture on these scaffolds. 

## Figures and Tables

**Figure 1 animals-12-00734-f001:**
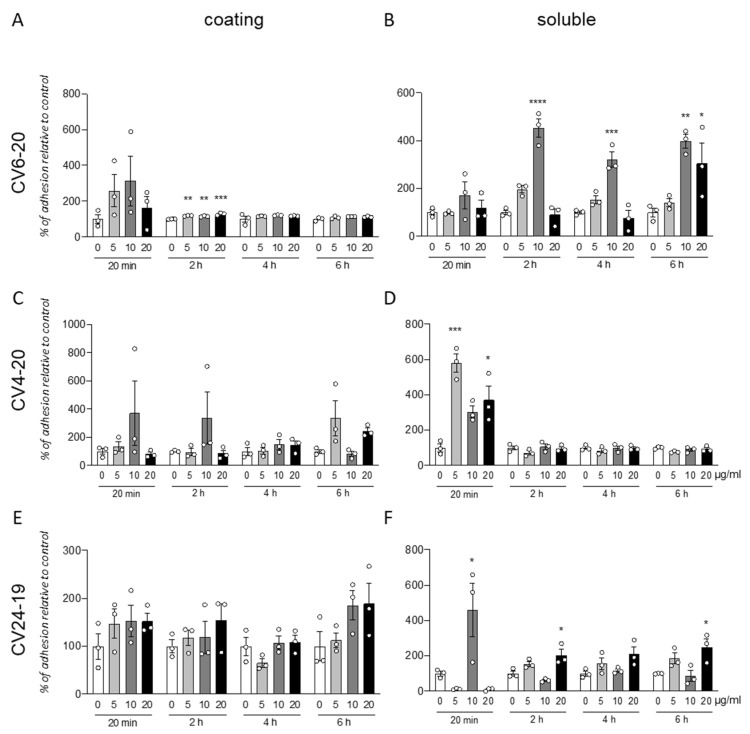
Cell adhesion assay for cells derived from adipose tissues. Graphs show the percentage of cell adhesion relative to control after 20 min, 2, 4, or 6 h from seeding. Cells were seeded on a surface coated with GM18 or vehicle (**A**,**C**,**E**) or exposed for 24 h to the molecule and afterward detached and seeded again (**B**,**D**,**F**). Results are expressed as a percentage of adhesion compared to the control (0 µg/mL; 100%) for each condition and each time point. Each column represents the mean value ± SEM. Statistical analysis. One-Way ANOVA within each condition and each time point, followed by Dunnett’s post-test. Asterisks represent the differences compared to the control group (* *p* < 0.05; ** *p* < 0.01; *** *p* < 0.001; **** *p* < 0.0001).

**Figure 2 animals-12-00734-f002:**
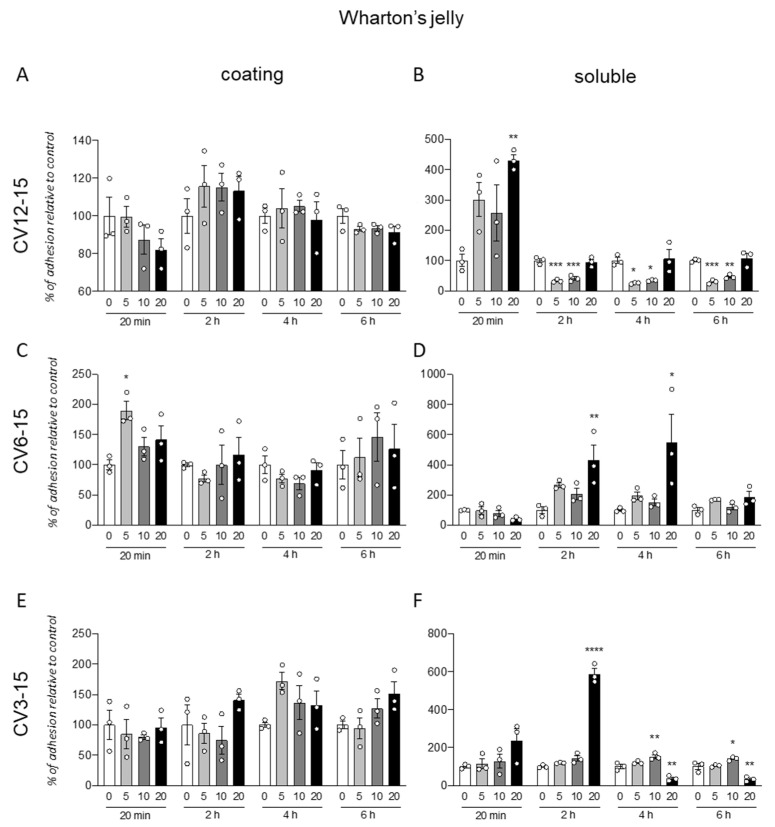
Cell adhesion assay for cells derived from Wharton’s jelly. Graphs show the percentage of cell adhesion relative to control after 20 min, 2, 4, or 6 h from seeding. Cells were seeded on a surface coated with GM18 or vehicle (**A**,**C**,**E**) or exposed for 24 h to the molecule and afterward detached and seeded again (**B**,**D**,**F**). Results are expressed as a percentage of adhesion compared to the control (0 µg/mL; 100%) for each condition and each time point. Each column represents the mean value ± SEM is shown. Statistical analysis. One-Way ANOVA within each condition and each time point, followed by Dunnett’s post-test. Asterisks represent the differences compared to the control group (* *p* < 0.05; ** *p* < 0.01; *** *p* < 0.001; **** *p* < 0.0001).

**Figure 3 animals-12-00734-f003:**
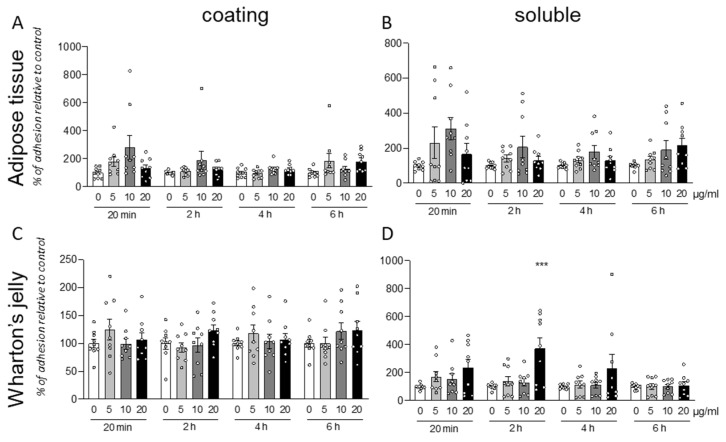
Cumulative analysis for cell adhesion assay. Graphs show the percentage of cell adhesion relative to control after 20 min, 2, 4, or 6 h from seeding. Cells were seeded on a surface coated with GM18 or vehicle (**A**,**C**) or exposed for 24 h to the molecule and afterward detached and seeded again (**B**,**D**). Results are expressed as a percentage of adhesion compared to the control (0 µg/mL; 100%) for each condition and each time point. Each column represents the mean value of all the tested wells for each group, pooling the results obtained from cells isolated from different animals ± SEM is shown. Statistical analysis. One-Way ANOVA within each condition and each time point, followed by Dunnett’s post-test. Asterisks represent the differences compared to the control group (*** *p* < 0.001).

**Figure 4 animals-12-00734-f004:**
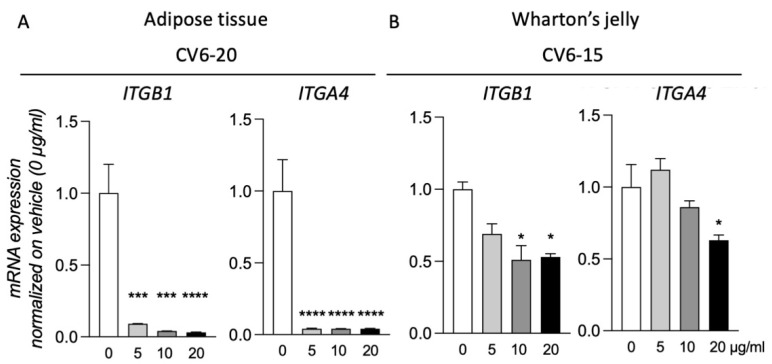
Gene expression regulation of the two target integrins. Graphs show the gene expression regulation of *ITGB1* and *ITGA4* genes in cells derived from CV6-20 adipose tissue (**A**) and CV6-15 Wharton’s jelly (**B**), treated for 24 h with different concentrations of GM18 (0, 5, 10, and 20 µg/mL). Columns represent the mean value + SEM. Statistical analysis. One-Way ANOVA followed by Dunnett’s post-test. Asterisks represent the differences compared to the control group (* *p* < 0.05; *** *p* < 0.001; **** *p* < 0.0001).

**Figure 5 animals-12-00734-f005:**
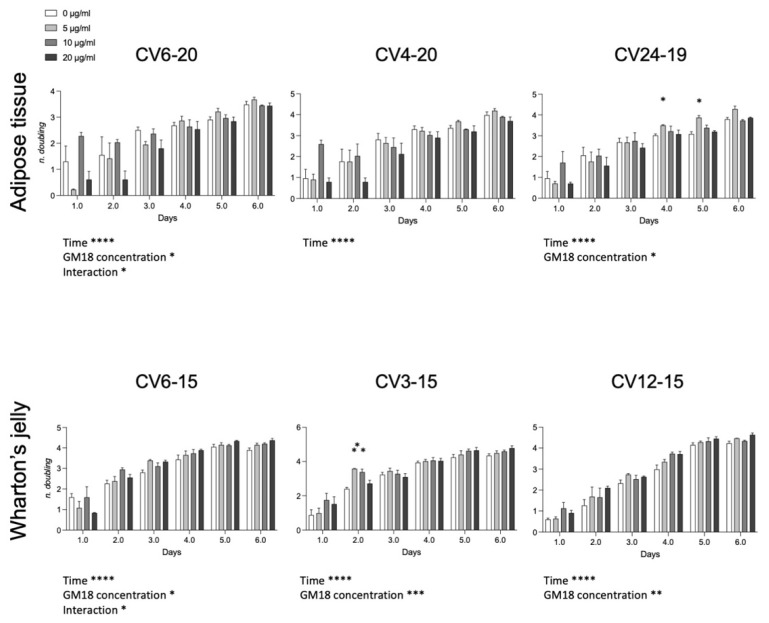
Effect of GM18 exposure on population doubling. Graphs show the number of doublings measured each day for 6 consecutive days, for cells isolated from adipose tissue and Wharton’s jelly exposed to GM18 at different concentrations (0, 5, 10, and 20 µg/mL). Statistical analysis. Two-Way ANOVA followed by Dunnett’s post-test. Asterisks represent the differences in the ANOVA parameters as indicated (* *p* < 0.05; ** *p* < 0.01; *** *p* < 0.001; **** *p* < 0.0001).

**Figure 6 animals-12-00734-f006:**
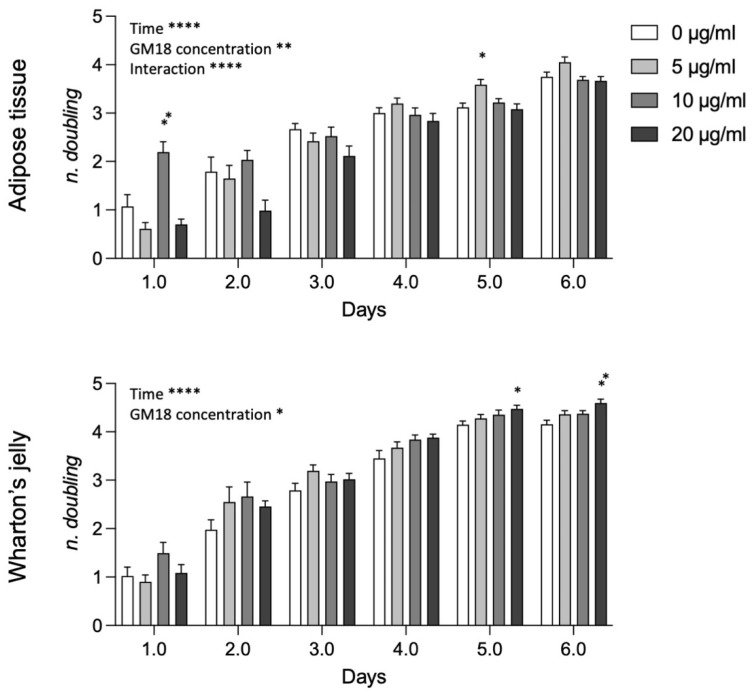
Effect of GM18 exposure on population doubling. Graphs show the number of doublings measured each day for six consecutive days, for cells isolated from adipose tissue and Wharton’s jelly exposed to GM18 at different concentrations (0, 5, 10, and 20 µg/mL). Statistical analysis. Two-Way ANOVA followed by Dunnett’s post-test. Asterisks represent the differences in the ANOVA parameters as indicated (* *p* < 0.05; ** *p* < 0.01; **** *p* < 0.0001).

**Figure 7 animals-12-00734-f007:**
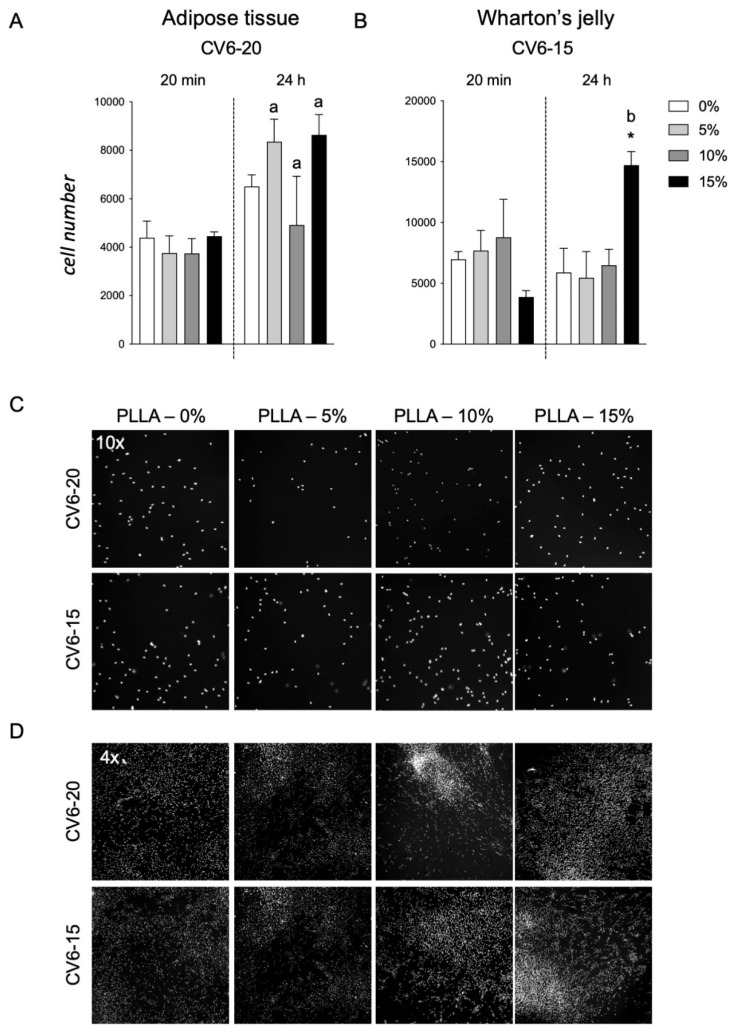
Cell adhesion assay for cells seeded on PLLA-GM18 scaffolds. Graphs show the number of counted cells per well in 20 min and 24 h after seeding the cells isolated from CV6-20 adipose tissue (**A**) and CV6-15 Wharton’s jelly (**B**) on PLLA scaffolds containing different concentrations of GM18 (0, 5, 10, and 15%). Representative images of Hoechst staining are included in the figure, showing the cultures at 20 min acquired with a 10× objective (**C**) and 24 h acquired a with a 4× objective (**D**) from the seeding. Statistical analysis. One-Way ANOVA followed by Dunnett’s post-test within the same time point; asterisk represents the differences compared to the control group seeded on PLLA 0% GM18 (* *p* < 0.05). Student’s *t*-test within the same experimental group between the two time points, letters represent the statistically significant differences (a = *p* < 0.05; b = *p* < 0.01).

**Table 1 animals-12-00734-t001:** Primer sequences for real-time PCR.

Gene	Accession Number	Primer Sequence
*GAPDH*	NM_001163856	F: 5′-GAT GCC CCA ATG TTT GTG A-3′
		R: 5′-AAG CAG GCA TGA TGT TCT GG-3′
*ITGA4*	XM_023622141	F: 5′-CAG ATG CCG GAT CGG AAA GA-3′
		R: 5′-GCC CAC AAG TCA CGA TGG AT-3′
*ITGB1*	XM_023631884	F: 5′-CCA AAT GGG ACA CGC AAG AA-3′
		R: 5′-GCA CAG CGA GTG CTC ATT TT-3′

## Data Availability

The data presented in this study are available within the article and Appendix A.

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
