# Peer review of "Peptide Mediated Adhesion to Beta-Lactam Ring of Equine Mesenchymal Stem Cells: A Pilot Study"

_animals, 2022, doi:10.3390/ani12060734_

Round 1

Reviewer 1 Report

The work of Merlo et al. addresses an exciting topic: the possibility of promoting cell adhesion and replication on scaffolds treated with integrin agonists (GM 18 in this particular case). The approach on the equine model was made based on results observed on MSCs of human origin. Unfortunately, the observed results are not adequate to assert the model’s effectiveness, as partially recognized by the authors. The authors draw conclusions with which I disagree: I believe their conclusions in some cases are not based on observed data. The variety and uncertainty of the results may be due to a strong individual and cellular variability, as they affirm. However, I believe that the number of donors used should have been greater precisely for this reason: this individual variability could have been confirmed excluding with greater certainty experimental/technical effects.

I believe that the discussion of the results should be modified, and the enormous variability and the impossibility - with the results here presented - of drawing positive conclusions on the use of G18 should be discussed more clearly.

Regarding the specific points of the work:

  1. English language should be checked. The manuscript is understandable, but there are many grammar or typing errors.
  2. Paragraph 2.6. The authors say “9mm Petri dishes” have been used. These measure in quite strange for standard Petri dishes. If 9mm is the diameter, the authors seeded 3200 cells per well (the surface of 9mm dish is 0.64 cm2). How can it count such a low number of cells using a haemocytometer (supposedly a Burker’s camera)? If the diameter is wrong, please rewrite the section. Furthermore, it is unclear how many dishes were counted for each experimental condition. I.e.: at each time point were counted 1 or, instead, 3 dishes? Overall the section is unclear.
  3. Paragraph 3.1 . Cells were pretreated and then re-plated. How many times were the experiments repeated for each donor? How many replicate wells were used? What does “multiwell plate” means: 24, 48, 96 wells plate?
  4. Figure 1,2 and 3. Please make sure that the y axis name is adjusted. The authors report “cell number per wells” when the results are expressed as a percentage, not as the number of counted cells. The same is for the figures’ legend. Regarding the results, it seems that sometimes attached cells increase following the treatment at first time points but then clearly decline at 6 hours (ex, Fig2 D,F): What could it mean in terms of cells safety for long time cultures? The authors should also discuss the rationale for using a pre-treatment before cell seeding. 
  5. The results described in Fig 1-3 demonstrate a substantial variation in the response between cells derived from different animals. Why did the authors not use a more significant number of donors to verify the meaning and robustness of the data?
  6. Lines 451-454. I disagree with the conclusions of the authors. Overall, the results do not sustain the assertion that MSCs are “positively affected”. AT-MSCs are not positively affected, while WJ cells are affected only at 2 hours-time point at one treatment concentration (fig 3, D). I do not think these findings suggest a possible GM18 application to improve equine MSCs adhesion ability.
  7. Authors write that population doubling number is affected (results and discussion section), but they never say if population doubling is positively or negatively affected. Unfortunately, figures are not clear, and for time points from 3 to 6 days, it is hard to think about a possible difference. Could the authors be more precise about this effect? At which time points are cells affected? To which extent?
  8. The same criticism could be addressed to PLLA scaffold experiments. The results are strongly negative for AT-MSCs, and the assertion that a longer time could be necessary to improve the outcome is only speculation: too large is the difference from the control. The authors should repeat the experiments appropriately to confirm their hypothesis. 
  9. I also doubt the convenience of using cells prepared from horses affected by colic, where endotoxiemia/toxiemia is quite frequent. The biological properties of MSCs could be somehow compromised.

Author Response

Author's notes to Reviewer 1 are in the attached file.

Reviewer 2 Report

The topic is interesting on the regenerative medicine point however, the study is only preliminary without sufficient details. although it forms a base for further studies.

The study does not discuss lacunas and the way out in clear way for further progression of the area.

Its applications remain to be shown

Author Response

Author's notes to Reviewer 2 are in the attached file.

Reviewer 3 Report

The manuscript titled: "Peptide mediated adhesion to beta-lactam ring of equine mesenchymal stem cells" presents an interesting approach to MSC cell culture, with sound results obtained and presented in the manuscript. The authors do a good job introducing the topic and presenting the results, as well as relating them to the newest findings of the literature. Moreover, they are aware of the limitations of their study and suggest the further steps necessary to apply its findings in further research. 

The only minor shortcoming of the manuscript are the occasional stylistic language mistakes, which could affect the reception of the manuscript by English language speakers. The author's work could greatly benefit from a professional proofing/editing service. 

Therefore, I recommend acceptance of the manuscript after a minor revision, relating particularly to the language of the paper.

Author Response

Author's notes to Reviewer 3 are in the attached file.

Round 2

Reviewer 1 Report

Reviewing the work of Merlo et al. improved the manuscript noticeably. The experimental scheme is better described, the results are presented more thoroughly, and their discussion is broader, recognizing the points of the work that need further study. I believe that the work is publishable in this form.